# Normalization of Muscle Strength Measurements in the Assessment of Cardiometabolic Risk Factors in Adolescents

**DOI:** 10.3390/ijerph18168428

**Published:** 2021-08-10

**Authors:** Tiago Rodrigues de Lima, Xuemei Sui, Diego Augusto Santos Silva

**Affiliations:** 1Research Center Kinanthropometry and Human Performance, Sports Center, Federal University of Santa Catarina, Florianopolis 88040-900, Brazil; tiagopersonaltrainer@gmail.com; 2Department of Exercise Science, Arnold School of Public Health, University of South Carolina, Columbia, SC 29208, USA; msui@mailbox.sc.edu

**Keywords:** adolescent health, cardiometabolic risk, cardiometabolic disorders, hand strength, physical fitness

## Abstract

Muscle strength (MS) has been associated with cardiometabolic risk factors (CMR) in adolescents, however, the impact attributed to body size in determining muscle strength or whether body size acts as a confounder in this relationship remains controversial. We investigated the association between absolute MS and MS normalized for body size with CMR in adolescents. This was a cross-sectional study comprising 351 adolescents (44.4% male; 16.6 ± 1.0 years) from Brazil. MS was assessed by handgrip and normalized for body weight, body mass index (BMI), height, and fat mass. CMR included obesity, high blood pressure, dyslipidemia, glucose imbalance, and high inflammation marker. When normalized for body weight, BMI, and fat mass, MS was inversely associated with the presence of two or more CMR among females. Absolute MS and MS normalized for height was directly associated with the presence of two or more CMR among males. This study suggests that MS normalized for body weight, BMI, and fat mass can be superior to absolute MS and MS normalized for height in representing lower CMR among females. Absolute MS and MS normalized for height were related to higher CMR among males.

## 1. Introduction

Muscle strength has been described as an important health marker in adolescents [1,2], in view of the inverse association with cardiometabolic risk factors analyzed individually (e.g., high body mass index—BMI, increased waist circumference, high systolic and diastolic blood pressure, high fasting blood glucose, increased levels of glycated hemoglobin, high levels of total cholesterol and triglycerides, high sensitive-C-reactive protein), or in terms of the simultaneous presence of two or more cardiometabolic risk factors (e.g., metabolic syndrome) [1,2,3]. However, even though studies have identified that muscle strength was inversely associated with cardiometabolic risk factors [1,2,3], absence of association or even contrary results (e.g., muscle strength directly associated with cardiometabolic risk factors) have been described [1,2,3].

The conflicting results for the association between muscle strength and cardiometabolic risk factors in adolescents can be related to the different strategies adopted by studies to consider body size in determining muscle strength (normalization for body size) [1,2,3]. While some studies normalized muscle strength for body mass [4,5], others adopted absolute muscle strength to investigate this interrelationship [6,7]. However, the best strategy to be adopted when considering body parameters in the expression of muscle strength values in adolescents is currently unknown [1,2]. Although muscle strength is directly impacted by muscle mass, fat mass, one of the components of muscle mass, can moderate the magnitude of the muscle mass/muscle strength interrelation [8]. In this sense, it is hypothesized that, when considering the values of muscle strength according to fat mass, in addition to minimizing confounding aspects attributed to the way of measurement, the quality of muscle strength, rather than quantity, is evidenced [9]. Additionally, based on the premise that adolescence is a period of changes in body structures [10], and that muscle mass plays an important role in maintenance of growth development [11], it is speculated that the expression of the values of muscle strength, according to the height, allows to include such development in the structure and body composition, and that the use of this technique can result in more reliable muscle strength values. Such an assumption is based on the fact that an increase in fat-free mass throughout puberty occurs similarly to an increase in height, at least in the shape of the growth curve [11].

Although some researchers have expressed the need to normalize muscle strength for body size [1,2], in studies that has had synthesized the results for the relationship between muscle strength and cardiometabolic risk factors [1,3], the normalization quantities of muscle strength measurements was restricted to body mass only. Thus, it is hypothesized that, when considering the size of the body, dimensions, and components of body composition in the expression of muscle strength, the values will be considered without the influence of confounding factors [12], and muscle strength normalized for body size will be a superior indicator for cardiometabolic risk factors in comparison with absolute muscle strength. Furthermore, considering that the genesis of cardiovascular diseases can begin during adolescence [13], and that the control and monitoring of risk factors associated with these conditions, as cardiometabolic risk factors, are configured as relevant strategies with the aim of preventing cardiovascular diseases [14], an investigation on the correlation of these factors, such as muscle strength measured by handgrip strength test (a simple, low-cost, and highly accurate method to measure muscle strength and as an indicator of the individual’s general health) [15] in adolescents is justified.

Thus, the present study aimed to investigate the association between absolute muscle strength and muscle strength normalized for body size (body weight, BMI, height, and fat mass) with cardiometabolic risk factors, which were analyzed individually or in terms of the presence of two or more cardiometabolic risk factors in adolescents. We hypothesized that muscle strength normalized for body weight, BMI, height, and fat mass is superior to absolute muscle strength as a predictor of cardiometabolic risk.

## 2. Materials and Methods

The present study aimed to investigate the association between absolute muscle strength and muscle strength normalized for body size (body weight, BMI, height and fat mass) with cardiometabolic risk factors in adolescents. In these investigations, the cardiometabolic risk factors were analyzed as individual factors or as grouped variables (either as the number of risk factors present in an individual or as combinations of risk factors). Additional information regarding the investigated population, sampling, analyzed variables and statistical plan can be identified throughout this section.

The data used in this cross-sectional analysis came from a school-based population study entitled “*Guia Brasileiro de Avaliação da Aptidão Física Relacionada à Saúde e Hábitos de Vida-Etapa II*” (Brazilian Guide for the Assessment of Health-Related Physical Fitness and Life Habits—Stage II). The aforementioned study was carried out in the second semester of 2019 and included representative sample of adolescents (14 to 19 years) enrolled in public high schools, living in São José, Southern Brazil. The aim of the macroproject was to investigate the relationship between health-related physical fitness indicators and clinical, blood, and lifestyle variables among adolescents. The study was approved by the Ethics Committee on Human Research of the Federal University of Santa Catarina (protocol nº 3.523.470). Adolescents under the age of 18 years had to present the Term of Assent and the Term of Free and Informed Consent signed by the guardians, or by the adolescent himself (age ≥ 18 years), and those with a physical disability that limited the performance of physical tests were not included in the research.

Sampling was performed in two stages: (1) stratified by public high schools (according to density-11 eligible public schools in São José); and (2) clustered by classes considering school shift and grade (186 high school classes-77.1% of students were on the day shift). Thus, considering that 5411 students (14–19 years old) were enrolled in the 2019 school year, a confidence level of 1.96 (95% confidence interval), a tolerable error of five percentage points, a prevalence of 50%, a 1.5 design effect, and the inclusion of an additional 20% to compensate for possible losses and refusals, and another 20% to control for potential confounders in the association analyses [16], the required sample size was 1233 students. In view of the lack of financial resources, a sub-sample of adolescents had their blood information collected (*n* = 371). Of this amount, 351 students had all information regarding cardiometabolic variables (clinical and blood), muscle strength and other variables investigated in the present study.

Since the present study used a set of information to examine issues different from those addressed in the broader previous study, the statistical power, which refers to the probability of rejecting a false null hypothesis (i.e., to determine that there is no difference when in fact there is) [17], was calculated, and values > 80% were considered adequate to prevent this error. In the present study, most of associations presented sufficient power to investigate association between cardiometabolic risk factors (either dichotomous and polytomous variables) and muscle strength indices. Full information regarding statistical power can be assessed in Appendix A. Data collection took place in the school environment in the second half of 2019, during the months of September to November.

The collection team was composed of undergraduate, master’s, and doctoral students. After collecting information, research members participated in training to standardize the application of questionnaires, conducting anthropometric assessments, applying physical tests, and assessing blood pressure. In these trainings, in addition to theoretical content, the evaluators were submitted to practical tasks, as well as for evaluation with the technical error of measurement (TEM) evaluator and inter-evaluator for anthropometric measurements according to the recommendations of literature [18]. To determine the TEM, ten subjects in the age group of the study and of different biotypes were randomly recruited to have their body mass, height, and waist circumference information measured (three times) by the interviewers. To calculate the TEM intra-rater and inter-rater, the measurements of the interviewers were compared with themselves and between them. In addition, results were compared with that of an anthropometristconsidered gold standard with experience in anthropometric measurements, and certified by the International Society for the Advancement of Kinanthropometry (ISAK). The maximum value of TEM inter (0.33%) and intra-evaluators (0.23%) was for the waist circumference measurement, which indicated an adequate level of the interviewers for anthropometric measurements [19]. Blood information was collected by nurses hired for the research.

Anthropometric indicators (height, waist circumference—WC, and body weight) were measured according to the recommendations of the literature [20]. The mean of 2 measurements for each of these indictors was considered for analysis. Height was measured with a Sanny^®^ stadiometer with tripod (Sanny: Sao Paulo, Brazil) with resolution in millimeters and maximum range of 213 cm. For height measurement, adolescents removed their shoes and headdresses. They were positioned in the middle of the equipment, standing motionless, erect with arms relaxed alongside, and leaning in the stadiometer ruler with the Frankfurt plane parallel to the base. WC (cm) was measured in the narrowest part of the trunk using an inextensible anthropometric tape (Sanny: Sao Paulo, Brazil) with a maximum capacity of 150 cm and a 1-mm resolution, and the body weight by using a G-tech^®^ digital scale (G-tech: Zhongshan, China) with resolution of 100 g and capacity of 150 kg. Among those adolescents aged 14–15 years, values equal to or greater than the 90th percentile for the WC [21] were used to define abdominal obesity. Adult criteria for abdominal obesity (WC > 80.0 cm for females and >90.0 cm for males) was adopted for those 16 years old or higher [21]. BMI was initially estimated as a continuous variable (kg/m^2^), and classified based on the cut-off points in z-scores proposed by the World Health Organization (WHO) [22] which defines obesity as ≥2 standard deviations.

The determination of fat mass was preceded by the calculation of the percentage of body fat (%BF), based on triceps and subscapular skinfold measurements, which were collected in millimeters [20]. Based on skinfold evaluation, %BF was estimated using the Lohman [23] predictive equation: %BF = 1.35 × (triceps + subscapular skinfold) − 0.012 × (triceps + subscapular skinfold)^2^ − constant (according to the sex, age, and ethnicity/race). In the present study, we considered the constants suggested by Pires-Neto and Petroski [24] who developed such constants to be used in Lohman’s [23] equation from a sample of Brazilian children and adolescents. To determine the ethnicity/race that were to be used in the equations, self-reported information by the students evaluated in the present study was used (white, brown, black, yellow and indigenous) according to Brazilian Standards [25]. After the %BF determination, the fat mass was obtained by the equation: fat mass = (body weight × %BF)/100.

Blood pressure measurements were performed using the oscillometric method through a calibrated Omron (Kyoto, Japan) electronic and digital device model HEM 742, with cuffs of appropriate size to fit the arms of adolescents. Blood pressure (systolic blood pressure—SBP; and diastolic blood pressure—DBP), both measured as mmHg, were collected according to the recommendations of the literature [26], i.e., they were measured twice, with rest time before and between measurements of at least 15 min. When the difference between measurements was greater than 10 mmHg for SBP or DBP, a third measurement was performed to replace the highest value. The mean of two measurements either for SBP or DBP was then calculated. High blood pressure (HBP) was identified as follows: (1) For the age group 14–17 years, SBP or DBP equal to or greater than values found in 95th percentile (P95) of the reference tables in “The fourth report on the diagnosis, evaluation, and treatment of high blood pressure in children and adolescents [27]”, which varies according to age and sex, adjusted for height percentile; and (2) for adolescents aged 18–19 years, values used for young adults, SBP ≥ 140 mmHg and/or DBP ≥ 90 mmHg [28] were used.

Venous blood samples were collected early in the morning after at least 8 h of fasting. Samples were stored and analyzed according to technical standards of the Laboratory of Clinical Analysis (University Hospital, Federal University of Santa Catarina—UFSC). Lipid profile—cholesterol (CHOL; mg/dL), triglycerides (TRG; mg/dL), LDL cholesterol (LDL-Chol; mg/dL), HDLcholesterol (HDL-Chol; mg/dL)—, fasting glucose (FBG; mg/dL), and fasting insulin levels (FIL; mU/L) were determined by the colorimetric test, while homeostatic model assessment for insulin resistance (HOMA-IR) was calculated as follows: HOMA-IR = (FBG × 0.0555 × FIL)/22.5 [29]. The high-sensitivity C-reactive protein (hs-CRP; mg/L) was determined by quantitative turbidimetry method. Values equal to or greater than 170 mg/dL, 90 mg/dL, and 110 mg/dL, and less than 45 mg/dL were used to define the higher values of CHOL, TRG, LDL-Chol, and the lower values of HDL-Chol, respectively [30]. Elevated FBG was defined as a value ≥ 100 mg/dL [31], while a value ≥ 3.16% was used to define adolescents with high concentrations of HOMA-IR [32]. Additionally, values equal to or greater than 3.0 mg/L were adopted to classify those with high hs-CRP [33].

Muscle strength was evaluated by using a handgrip dynamometer (Saehan manual dynamometer: Seoul, Korea) with the individual in a standing position and their arms extended straight down to the side [34]. The equipment was located between the distal phalanges and the palm. After that, the subject was asked to take inspiration and maximum expiration, followed by the greatest pressure with the hand on the equipment after a verbal encouragement. The test was performed on both hands alternately, three times, and the best result of each hand was summed and recorded in kilogram/force (kgf). In this study, the results obtained were used in accordance with the recommendations of the International System of Units (SI), being expressed in Newtons-N (kgf × 9.80). The handgrip strength test was chosen because handgrip strength levels have been strongly correlated with total muscle strength (correlation coefficient 0.736 to 0.890) in children, adolescents, and adults [15].

Two different outcomes were considered for this study: (1) individual cardiometabolic risk factors, including obesity (defined by WC or BMI), HBP, dyslipidemia (high values of CHOL, TRG, LDL-Chol or low values of HDL-Chol) [20], glucose imbalance (high FBG or high HOMA-IR), and high inflammation marker (elevated hs-CRP); and (2) the number of cardiometabolic risk factors within the same individual (0, 1, 2, 3, 4, or 5 cardiometabolic risk factors). Since only 13.9% of the sample had three or more cardiometabolic risk factors, we decided to group them into one category. Thus, for the purpose of analysis, adolescents investigated in this study were classified as presenting “0”, “1”, or “2 or more” cardiometabolic risk factors.

Since there is no standard measurement for muscle strength, and considering the evident effect of the body size on muscle strength values in children and adolescents [1,2,3,9,12], in the present study muscle strength was expressed through distinct indices considering some issues regarding the body size: (a) absolute muscle strength (sum of the values obtained in both hands-N); (b) muscle strength normalized for bodyweight (sum of the value obtained in both hands relative and expressed by body weight-N/kg); (c) muscle strength normalized for BMI {sum of the value obtained in both hands and expressed by BMI-[N/(kg/m^2^)]}; (d) muscle strength normalized for fat mass (sum of the value obtained in both hands and expressed by fat mass ‒ N/fat mass); (e) and muscle strength normalized for height (sum of the value obtained in both hands and expressed by the height—N/height) (Figure 1).

Sociodemographic, lifestyle, and sexual maturation were included as control variables due to the relationship with cardiometabolic variables and muscle strength [1,2,3,35]. Sex (male/female), age (collected in years), and socioeconomic levels were aspects of a questionnaire to evaluate the purchasing power—from “E” (lower purchasing power) to “A” (higher purchasing power)—of the adolescents’ families [36].

Physical activity level was assessed by the following question of a validated questionnaire Brazilian population [37]: during the past 7 days, how many days were you physically active for at least 60 min a day (consider the time you spent in any kind of physical activity that increased your heart rate and made your breathing faster for some time)? Adolescents who responded to perform physical activity for at least 60 min, seven days a week, were considered as meeting recommendations for physical activity [38]. In addition, individuals aged 18 years or over were considered as meeting recommendations for physical activity when performed at least 150 min of moderate-intensity physical activity throughout the week or at least 75 min of vigorous-intensity physical activity throughout the week, or an equivalent combination of moderate-and vigorous-intensity activity [38].

Information regarding muscle strengthening exercises were investigated by the following question [39]: “during a typical week, how many days do you exercise to improve the tone and strength of your muscles such as weight training and gymnastics?”. Adolescents who responded that they engaged in muscle strength exercises on three or more days per week were considered as meeting recommendations for muscle strength exercises [38]. Additionally, schoolchildren aged 18 years or higher were considered as meeting recommendations for performance of muscle strength improvement exercises when they performed muscle strength exercises at least two times per week [38].

The questions regarding eating habits and smoking came from a translated and validated questionnaire for the Brazilian population [40]. Information regarding eating habits was collected by questions related to a typical week: “do you have a balanced diet?”. Response options for this question were: hardly ever; rarely; sometimes; with relative frequency; often. A balanced diet was composed of cereals and grains (5 to 12 servings per day); fruits and vegetables (5 to 10 servings per day); meats and meat products (2 to 3 servings a day); and milk and dairy products (3 to 4 servings up to 16 years and 2 to 4 portions over 16 years) [34]. Response options (almost never, rarely, and sometimes) were considered a less frequent response. Response options with relative frequency and often were considered a frequent response. Smoking was assessed by the follow question: “do you smoke cigarettes?”. Individuals who responded “never smoked” were considered negative for smoking, and those that responded more than 10 per day; 1 to 10 per day; none in the last six months; none last year, were considered as positive for smoking.

The question regarding alcohol use was as follows [37]: “during the last 30 days, how many days did you drink five or more alcoholic drinks in a single occasion? (A dose corresponding to one can of beer, a glass of wine, a shot of whiskey, rum, rum, vodka, etc.)”; those who answered at least once were considered positive response for alcohol consumption.

Sexual maturation was assessed according to Tanner’s criteria [10] through the use of figures indicating maturational development adopted in a sample of Brazilian schoolchildren [41]. The stages of sexual maturation are indicated by self-assessment (figures) of breast development and pubic hair (female), and genital development and pubic hair (male). Prior to the application of questions related to sexual maturation, the adolescents were separated into different rooms according to gender, and additional information regarding the interpretation of the figures, indicating the maturational development, was provided by a member of the research of the same sex as the individuals evaluated. Stage 1 represents the prepubertal stage, stages 2–4 represent puberty, and stage 5, i.e., the post-pubertal stage. In the present study, adolescents were classified as prepubertal, pubertal, and post-pubertal [10,41].

Mean and standard deviation were used to describe normally distributed variables, and median and interquartile range (pp. 25–75) were used for non-normal variables. Categorical variables were presented as percentages (%). Depending on the nature of the investigated variables, the chi-square test, *t*-test for independent samples, or the Mann–Whitney test were used to identify possible differences according to sex.

Initially, logistic regression was used to investigate the association between individual cardiometabolic risk factors with muscle strength indices. Results were presented as natural logarithm of odds ratio (lnOR) along with 95% confidence intervals (CIs), the borders of which were presented in square brackets—[CI_lower; CI_upper]. Multinomial logistic regression was adopted to test the association between number of cardiometabolic risk factors (“0” risk factor as reference) with muscle strength. Results were presented as lnOR with their 95% CIs, the borders of which were presented in square brackets—[CI_lower; CI_upper]. Additionally, such results from multinomial logistic regression were also presented graphically as predicted probabilities according to percentiles of muscle strength (10th, 25th, 50th, 75th, and 90th). In view of the possible effect of sex and age on muscle strength [1,2,3,26], interactions between these factors in the association with cardiometabolic risk factors (individual and grouped in terms of presence of two or more cardiovascular risk factors) were tested in the regression models. For most of tested analysis, interaction between muscle strength and sex (*p* value < 0.05) was verified. Thus, results were stratified by sex in the present study. All analyses were adjusted for all possible confounders (sociodemographic and lifestyle variables) [1,2,3,26], regardless of their level of statistical significance in the association with the outcomes.

Data analysis was conducted in the statistical software Stata 16.0 (StataCorp LP, College Station, TX, USA), considering sampling weights and the survey design.

## 3. Results

A total of 351 adolescents (age, 16.6 ± 1.0 years; male, *n* = 155; female, *n* = 196) with full information for all investigated outcomes were assessed in the present study. Descriptive characteristics of the sample and deep information regarding individual cardiovascular risk factors or the number of cardiovascular risk factors according to sex can be assessed in Appendix A.

Among males, muscle strength normalized for body weight was inversely associated with obesity—lnOR: −4.605 [−4.803; –3.912], glucose imbalance—lnOR: −3.219 [−4.605; –2.040] and high inflammation marker—lnOR: −2.040 [−3.912; –0.020]. When considering the BMI in the normalization of muscle strength, muscle strength was inversely associated with obesity—lnOR: −1.609 [−2.120; −1.171] and high blood pressure—lnOR: −0.635 [−1.108; −0.162]. Additionally, muscle strength normalized for fat mass was inversely associated with obesity—lnOR: −1.204 [−1.897; −0.511] (Table 1).

In the investigation of the relationship between muscle strength and cardiometabolic risk factors among females, muscle strength normalized for body weight was inversely associated with obesity—lnOR: −4.605 [−4.764; −3.912], dyslipidemia—lnOR: −2.408 [−2.996; −1.714], and glucose imbalance—lnOR: −3.219 [−3.506; −2.995]. Similarly, muscle strength normalized for BMI was inversely related to obesity lnOR: −1.897 [−3.219; −0.713], dyslipidemia lnOR: −0.755 [−1.514; −0.020], glucose imbalance—lnOR: −1.204 [−1.427; −0.994], and high inflammation marker—lnOR: −1.514 [−2.813; −0.174]. Furthermore, muscle strength normalized for fat mass was inversely associated with obesity—lnOR: −2.302 [−3.912; −0.821] and glucose imbalance—lnOR: −0.314 [−0.446; −0.186] (Table 2).

According to multinomial logistic regression, when compared with those without cardiometabolic risk factors, absolute muscle strength—lnOR: 0.029 [0.010; 0.048] and muscle strength normalized for height—lnOR: 0.048 [0.010; 0.086] were directly associated with the presence of two or more cardiometabolic risk factors among males (Table 3). Among females, muscle strength normalized for body weight—lnOR: −3.912 [−4.609; −0.061], BMI—lnOR: −1.560 [−2.813; −0.342], and fat mass—lnOR: −1.021 [−1.714; −0.301] were inversely associated with the presence of two or more cardiometabolic risk factors in the comparison with those without cardiometabolic risk factors (Table 3). Additionally, results from multinomial logistic regression were also presented graphically as predicted probabilities according to percentiles of muscle strength indices (10th, 25th, 50th, 75th, and 90th) for males (Figure 2) and females (Figure 3), whose specific values for each estimative can be accessed in full in Appendix A.

## 4. Discussion

The main purpose of this research was to investigate the association between absolute muscle strength and muscle strength normalized for body size (body weight, BMI, height, and fat mass) with cardiometabolic risk factors. The results of the present study indicated that among males, absolute muscle strength and muscle strength normalized for height were related to higher cardiometabolic risk. In addition, muscle strength according to body weight, BMI, and fat mass was inversely associated with the simultaneous presence of two or more cardiometabolic risk factors among females.

Concomitant with the increase in body dimensions, the magnitude of the relationship between muscle strength and body size tends to increase, possibly due to the increase in muscle mass and physiological muscle cross-sectional area (proportional to height and body weight) [12]. This aspect is particularly important to be considered when normalizing muscle strength, since the physiological cross-sectional area of the muscle and the recruitment ability of the motor units will determine the magnitude of muscle strength generation [42]. This relationship exerted by body size on muscle strength seems to be even more determinant depending on the type of test used to assess muscle strength [12]. While some tests to assess muscle strength require the need for propulsion (e.g., jumps) or body support (e.g., pull-ups), where in fact the direct impact that body size has on performance is identified, tests such as the handgrip strength, in which the need to jump or support the body is not required, are also directly impacted by the size of the body. This is because handgrip strength is closely related to body mass, and the values of muscle strength identified by handgrip strength will be higher among taller and heavier individuals, compared to those lower and lighter [12]. In addition to the impact of body size on the values obtained by the handgrip test, there is a growing body of evidence suggesting that muscle strength normalized for body-volume-related indexes (e.g., body weight, fat mass) may provide more reliable measures of muscle strength compared to those derived from absolute muscle strength values or muscle strength normalized for body-length-related indexes (e.g., height) when assessed by handgrip in the association with cardiometabolic indicators [1,2]. This is because, when using absolute values or restricting the expression of muscle strength values to body-length-related indices, body composition components, similarly important for determining muscle strength and/or directly associated with cardiometabolic indicators, such as fat mass, can confuse the direction of associations [8]. Thus, considering that body weight is also an important factor associated with increased cardiometabolic risk, and that body fat (further increases in body size are predominantly based on gain in fat tissue) is the link for clustering of cardiometabolic risk factors [43], do not consider body-volume-related indexes when normalizing muscle strength for body size in the investigation of the association between muscle strength assessed by handgrip with cardiometabolic risk factors is likely to provide unreliable results (especially for adolescents with large body mass) [8]. Thus, it is hypothesized that the disagreement between results for the direct association between absolute muscle strength and muscle strength normalized for height may be related to the use, or not, of body-volume-related indexes (i.e., volume of any body segment) as a strategy to consider body size when normalizing muscle strength obtained from a handgrip test.

The results identified in the present study regarding the inverse association between muscle strength (normalized for body weight, BMI, and fat mass) and the simultaneous presence of multiple cardiometabolic risk factors are in accordance with the literature [3,44]. In the study conducted in South Korea (*n* = 1050 schoolchildren; 574 boys, 476 girls; age 10–18 years), the presence of multiple cardiometabolic risk factors in the same individual (metabolic syndrome) was inversely associated with muscle strength (assessed by handgrip), expressed according to body weight in males and females [44]. Similarly, a systematic review study identified that muscle strength was inversely associated with the simultaneous presence of two or more cardiometabolic risk factors in adolescents (metabolic syndrome and individual components of the metabolic syndrome analyzed as a clustering of cardiometabolic risk factors, e.g., obesity, high blood pressure, dyslipidemia, and dysglycemia) [3]. However, it is noteworthy that the results of the aforementioned review study [3] did not consider strategies adopted in the expression of muscle strength values when synthesizing the body of evidence. Despite the mechanisms that support the relationship between muscle strength and the presence of two or more cardiometabolic variables in the same individual are not established [1,2,3], muscle strength is directly related to skeletal muscle [45], which in turn is the main site of insulin-mediated glucose uptake and accounts for approximately 75% of the body’s total uptake of insulin-stimulated glucose [46]. Imbalances in relation to insulin metabolism (insulin resistance) precede the development of multiple single and clustered cardiometabolic risk factors which precede the development of cardiometabolic diseases [46]. Thus, it is possible that higher levels of muscle strength (and consequently greater muscle mass and muscle quality) [47] contribute to the improvement of insulin metabolism and lower susceptibility to an increased risk (two or more cardiometabolic risk factors) of cardiometabolic diseases [46,47]. With regard to the restricted results for females, in this study, females had a higher amount of body fat (Appendix A), which in turn is associated with difficulties in improving muscle strength [47] and lower increase in fat-free muscle mass for women, but not for males [48]. Thus, it is speculated that the increase in muscle strength levels in females may reflect superior cardiometabolic benefits with respect to the presence of numerous cardiometabolic risk factor in the same individual.

The present study presents strengths that must be highlighted, including the use of objectively measured information regarding the outcome and exposure, and the adoption of several muscle strength indices in the association with cardiometabolic risk factors, analyzed individually or in terms of the number of risk factors. However, some limitations should be described. (1) Although the obtaining of muscle strength was determined by the sum of the values of muscular strength identified by both the dominant hand and the non-dominant hand, the inclusion of the values obtained by the non-dominant hand may have reduced the maximum strength obtained and impacted the relationship with cardiometabolic risk factor. In this same sense, although the assessment of muscle strength through handgrip test is a valid indicator of general muscle strength [15], it is possible that aspects related to the activities in which the evaluated individuals are engaged (i.e., activities requiring the efforts of the upper limbs-especially those in which there is a need to overcome resistance) can contribute to obtaining values with lower accuracy. (2) The sample was not big enough to report individual results from those with three, four, or five cardiometabolic risk factors. (3) The adoption of only one instrument to assess muscle strength, which limits the investigation of the total dimensions of muscle strength (e.g., endurance, power), is another limitation of this study. (4) The cross-sectional design does not allow assessment of muscle strength impact cardiometabolic risk factors in male and female adolescents over time.

## 5. Conclusions

The results of the present study indicated an inverse association between muscle strength normalized for body weight, BMI, or fat mass, and the simultaneous presence of two or more cardiometabolic risk factors in females, suggesting that when normalized for body-volume-related indexes, muscle strength can be superior to absolute muscle strength and muscle strength normalized for height in representing lower cardiometabolic risk for this subgroup of adolescents. Among males, absolute muscle strength and muscle strength normalized for height were associated with higher cardiometabolic risk. Longitudinal studies and clinical trials are needed to discover whether muscle strength can be a convincing predictor for the occurrence of several cardiometabolic risk factors in the same female individuals, and to confirm the results regarding the direct relationship of muscle strength with increased cardiometabolic risk when body volume-related indices are not considered in the expression of muscle strength values.

## Figures and Tables

**Figure 1 ijerph-18-08428-f001:**
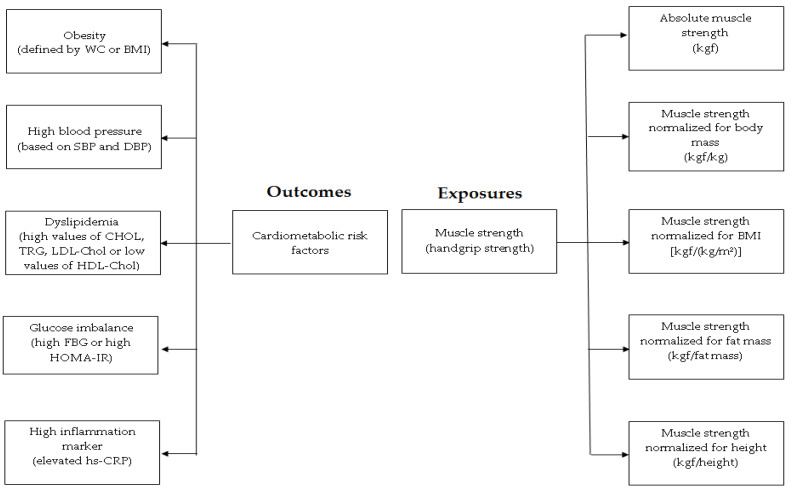
Diagram of the investigated variables. WC: waist circumference; BMI: body mass index; SBP: systolic blood pressure; DBP: diastolic blood pressure; CHOL: cholesterol; TRG: triglycerides; LDL-Chol: low-density lipoprotein cholesterol; HDL-Chol: high-density lipoprotein cholesterol; FBG: fasting blood glucose; HOMA-IR: homeostatic model assessment for insulin resistance hs-CRP: high-sensitivity C-reactive protein.

**Figure 2 ijerph-18-08428-f002:**
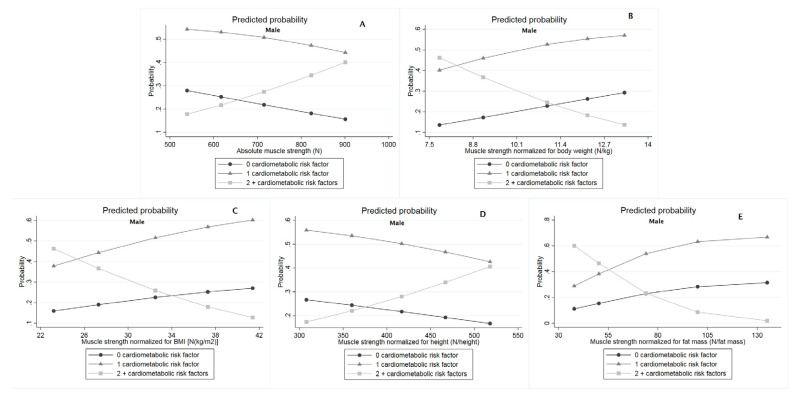
Predicted probability from a multinomial logistic model for the adjusted relationship between muscle strength indices with cardiometabolic risk factors among males. (**A**) Absolute muscle strength (N); (**B**) Muscle strength normalized for body weight (N/kg); (**C**) Muscle strength normalized for BMI [N/(kg/m^2^)]; (**D**) Muscle strength normalized for height (N/height); (**E**) Muscle strength normalized for fat mass (N/fat mass).

**Figure 3 ijerph-18-08428-f003:**
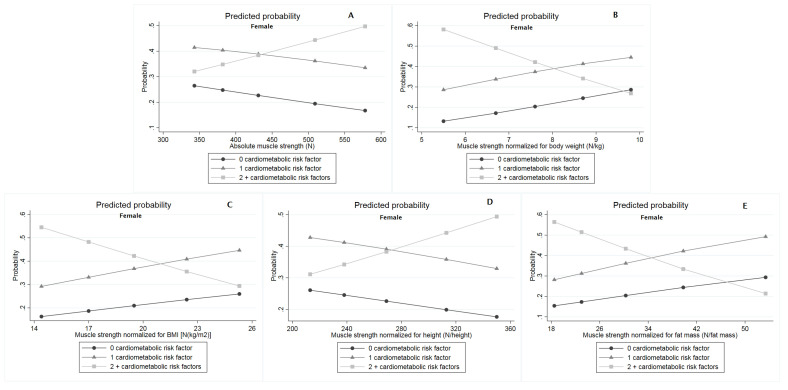
Predicted probability from multinomial logistic model for the adjusted relationship between muscle strength indices with cardiometabolic risk factors among females. (**A**) Absolute muscle strength (N); (**B**) Muscle strength normalized for body weight (N/kg); (**C**) Muscle strength normalized for BMI [N/(kg/m^2^)]; (**D**) Muscle strength normalized for height (N/height); (**E**) Muscle strength normalized for fat mass (N/fat mass).

**Table 1 ijerph-18-08428-t001:** Adjusted ^a^ association from binary logistic regression between muscle strength and cardiometabolic risk factors among males.

Outcomes	Absolute Muscle Strength (N)		Muscle Strength Normalized for Body Weight (N/kg)		Muscle Strength Normalized for BMI [N/(kg/m^2^)]		Muscle Strength Normalized for Height (N/Height)		Muscle Strength Normalized for Fat Mass (N/Fat Mass)	
	lnOR	(95% CI)	lnOR	(95% CI)	lnOR	(95% CI)	lnOR	(95% CI)	lnOR	(95% CI)
Obesity ^b^										
Yes	0.011	(−0.051; 0.086)	−4.605 *	(−4.803; −3.912)	−1.609 *	(−2.120; −1.171)	0.487	(−0.051; 0.157)	−1.204 *	(−1.897; −0.511)
Dyslipidemia ^b^										
Yes	0.019	(−0.010; 0.048)	0.157	(−2.525; 2.827)	0.285	(−0.174; 0.742)	0.029	(−0.040; 0.104)	0.003	(−0.128; 0.122)
Glucose Imbalance ^b^										
Yes	0.009	(−0.116; 0.113)	−3.219 *	(−4.605; −2.040)	−0.844	(−3.912; 2.131)	0.007	(−0.198; 0.199)	−0.162	(−0.527; 0.215)
High blood pressure ^b^										
Yes	0.020	(−0.030; 0.077)	−2.525	(−4.711; 0.342)	−0.635*	(−1.108; −0.162)	0.029	(−0.020; 0.086)	−0.174	(−0.693; 0.343)
High Inflammation Marker ^b^										
Yes	−0.030	(−0.072; 0.010)	−2.040 *	(−3.912; −0.020)	−0.616	(−1.272; 0.048)	−0.051	(−0.116; 0.010)	−0.128	(−0.713; 0.470)

N: Newtons; lnOR: natural logarithm of odds ratio; CI: confidence interval; BMI: body mass index. *: *p* value for association < 0.05. ^a^: Results adjusted for age, socioeconomic level, physical activity, muscle strength exercise, eating habits, smoking, excess alcohol use and maturational status; ^b^: Category “No” as reference.

**Table 2 ijerph-18-08428-t002:** Adjusted ^a^ association from binary logistic regression between muscle strength and cardiometabolic risk factors among females.

Outcomes	Absolute Muscle Strength (N)		Muscle Strength Normalized for Body Weight (N/kg)		Muscle Strength Normalized for BMI [N/(kg/m^2^)]		Muscle Strength Normalized for Height (N/Height)		Muscle Strength Normalized for Fat Mass (N/Fat Mass)	
	lnOR	(95% CI)	lnOR	(95% CI)	lnOR	(95% CI)	lnOR	(95% CI)	lnOR	(95% CI)
Obesity ^b^										
Yes	0.058	(−0.020; 0.113)	−4.605 *	(−4.764; −3.912)	−1.897 *	(−3.219; −0.713)	0.104	(−0.20; 0.231)	−2.302 *	(−3.912; −0.821)
Dyslipidemia ^b^										
Yes	0.001	(−0.072; 0.077)	−2.408 *	(−2.996; −1.714)	−0.755 *	(−1.514; −0.020)	0.004	(−0.094; 0.095)	−0.287	(−0.654; 0.095)
Glucose Imbalance ^b^										
Yes	−0.030	(−0.151; 0.086)	−3.219 *	(−3.506; −2.995)	−1.204 *	(−1.427; −0.994)	−0.051	(−0.248; 0.140)	−0.314 *	(−0.446; −0.186)
High blood pressure ^b^										
Yes	0.029	(−0.020; 0.077)	0.378	(−4.606; 1.795)	0.343	(−1.660; 3.040)	0.039	(−0.010; 0.095)	−0.116	(−1.203; 0.947)
High Inflammation Marker ^b^										
Yes	0.010	(−0.010; 0.029)	−3.912	(−4.599; 0.131)	−1.514 *	(−2.813; −0.174)	0.029	(−0.010; 0.058)	−0.654	(−1.514; 0.207)

N: Newtons; lnOR: natural logarithm of odds ratio; CI: confidence interval; BMI: body mass index. *: *p* value for association < 0.05. ^a^: Results adjusted for age, socioeconomic level, physical activity, muscle strength exercise, eating habits, smoking, excess alcohol use and maturational status; ^b^: Category “No” as reference.

**Table 3 ijerph-18-08428-t003:** Adjusted ^a^ association from multinomial logistic regression between muscle strength and number of cardiometabolic risk factors according to sex.

Outcomes	Absolute Muscle Strength (N)		Muscle Strength Normalized for Body Weight (N/kg)		Muscle Strength Normalized for BMI [N/(kg/m^2^)]		Muscle Strength Normalized for Height (N/Height)		Muscle Strength Normalized for Fat Mass (N/Fat Mass)	
	lnOR	(95% CI)	lnOR	(95% CI)	lnOR	(95% CI)	lnOR	(95% CI)	lnOR	(95% CI)
Male										
Number of cardiometabolic risk factors										
0 (reference)										
1	0.010	(−0.10; 0.029)	0.190	(−3.912; 1.982)	0.322	(−0.580; 1.215)	0.020	(−0.020; 0.049)	0.002	(−0.248; 0.262)
2+	0.029 *	(0.010; 0.048)	−4.605	(−4.723; 1.572)	−0.654	(−1.660; 0.336)	0.048 *	(0.010; 0.086)	−0.261	(−1.078; 0.425)
Female										
Number of cardiometabolic risk factors										
0 (reference)										
1	0.008	(−0.010; 0.010)	−1.832	(−4.605; 2.297)	−0.562	(−2.040; 0.883)	−0.010	(−0.030; 0.010)	−0.151	(−0.342; 0.029)
2+	0.039	(−0.072; 0.157)	−3.912 *	(−4.609; −0.061)	−1.560 *	(−2.813; −0.342)	0.058	(−0.127; 0.254)	−1.021 *	(−1.714; −0.301)

N: Newtons; lnOR: natural logarithm of odds ratio; CI: confidence interval; BMI: body mass index. *: *p* value for association < 0.05; ^a^: Results adjusted for age, socioeconomic level, physical activity, muscle strength exercise, eating habits, smoking, excess alcohol use, and maturational status.

## Data Availability

The data used in the present study, as well as the commands used to obtain the results described, are available upon request to the authors.

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
