# Peer review of "Normalization of Muscle Strength Measurements in the Assessment of Cardiometabolic Risk Factors in Adolescents"

_ijerph, 2021, doi:10.3390/ijerph18168428_

Round 1
Reviewer 1 Report
-----( General comments: )-----
This study needs a lot more work from the start, because scientific problem is not stated and novelty of the research is not presented. The work concerns normalization of muscle strength (standardised coefficients) in studies on cardiovascular risk factors, but those specific terms do not appear in the paper. Likewise, terms such as „normalization” or „risk factor” are completely absent in the work. Those lacks are indicative of poor recon on the language commonly used to describe the subject, which should have been an important part of the translation process.
-----( Title )-----
The title is not informative as it poorly introduces typical terms and as the study does not provide any new knowledge.
-----( Abstract )-----
Instead of a background, a purpose of the study is presented. A lack in current knowledge, which could be completed by the results, is not stated.
The authors write that muscle strength is related to Adverse Cardiometabolic Conditions, but they did not state the direction of this relationship.
Method and Results do not provide clear main information. Conclusions are confusing, do not refer to men and do not explain what should be understood by „sensible strategy”.
-----( Introduction )-----
The introduction does not provide any rationale or justification as to why the study should be conducted. There is no critical evaluation, no explanations on what is lacking among presented background information and which hypothesis can compensate for the lack. How is “the possible impact of the body size and its components on these associations” highlighted by the presented study? Normalisation of muscle strength is not a new subject. Is it lacking in existing studies about cardiovascular risk factors? Or perhaps the method of strength measurement itself is relevant here? What is its utility in assessment of the diagnosed feature and what is this feature, to begin with?
To „investigate association” is a method, not a scientific goal related to hypothesis. What analytical procedure was chosen to verify what is „more reliable”, and is it sufficient to prove what is more diagnostic?
The authors wrote that "the beneficial effect attributed to muscle strength on adverse cardiometabolic conditions in adolescents does not seem to be a consensus", but they did not write what this benefit is.
The authors did not present the research problem again. The problem must not be showing the relationship between indicators - the knowledge of these relationships must serve the scientific problem that is being solved in the study.
-----( Method )-----
Since this section is quite extensive, at its beginning a general plan of the investigation should be included. Meanwhile an attempt of its presentation can be found only in the Abstract.
What was the enrollment of a few thousands of students described for, if it did not affect the randomisation? The criteria of selecting 351 subjects to conclusively participate in the study was not specified, so this part of description is irrelevant.
The exclusion of subjects with three risk factors was decided, while their case would be the most interesting. After all, it is possible to create a strength characteristic of a group of 40.
A diagram organizing the examined indices would improve the reception of the text.
The least methodological information concerns strength assessment.
Since as a minimum to get required statistical confidence, the authors estimated the number of 1233 participants, why was the study continued with the group size of 351 (more than 3 times less than this minimum)? What had changed in assumptions or research purposes that this had become possible?
If in the data sets there are those whose distribution differs from normal, then the non-parametric method of description should be used for all data sets - different description methods ought not to be used for individual data sets in the same test.
The authors wrote: „Categorical variables were presented as percentages (%)” - If the data is expressed in quantitative units, they are no longer categorical. They can at most describe the categorical data in a quantitative manner (i.e. statistical frequency).
-----( Results )-----
There is no clear result that could indicate „better reliability” of the normalization method. The results are presented in a very descriptive manner, without applying the order of priority, which, given their number, notably obstructs the reception. Captions below the tables do not indicate the meaning of results. Tables are very descriptive, do not simplify the insight. What exactly differs men and women in terms of received relationships and how does it translate into the interpretation of results?
The data of the research group has already been described in methodology. There is no need to present this description again.
The method of presenting results in the tables requires a thorough refinement to eliminate empty spaces and thus reduce the size of tables and increase their readability.
There is no need to quote the results presented in the table again in the text - the Authors only have to comment on them simply.
Please express force values in SI units (N instead of kgf)
-----( Discussion )-----
A lot of very important information, which can be helpful in the introduction, so they should be shifted into this section. Interpretation of results leading to conclusion is lacking.
Author Response
Response to the Reviewers' Comments
Article number: ijerph-1223408
Article title: Association between muscle strength expressions and adverse cardiometabolic conditions in adolescents
Article authors: de Lima, Sui and Silva.
We are so grateful for the thoughtful reviews provided by the referees. We have tried to carefully address each recommendation. We believe that our manuscript is much stronger as a result of the significant modifications outlined below. We also thank the editoral staff and peer-reviewers for their contributions.
The modified text is written in red in the revised manuscript. In addition to the specific responses to the referees shown below, we performed some general editing of the text in order to maintain the standards required for the journal

Reviewer 2 Report
This is a well conducted study however one limitation (not mentioned) when considering hand strength as a proxy for overall muscle strength is whether the individual was involved in predominantly upper limb activities compared to lower limb activities ie the difference in hand strength between those that play tennis (or other predominant upper limb sports) and those that don't.
Comment on the role of muscle "bulk" measurement should also be made ie role of MRI (or perhaps US) to measure muscle bulk.
Otherwise happy with the overall results and discussion.
Author Response

(The authors gave the same response as above.)

Reviewer 3 Report
Thank you for an interesting article with the aim of investigating the association of muscle strength expressions with adverse cardiometabolic conditions among adolescents.
Abstract
Background: “We investigated the association of muscle strength expressions with adverse cardiometabolic conditions in adolescents.” I believe this does not explain the background for investigating the purposed research question. Please add this information.
Results: Please explain why the results are highlighted for females only (see also comments under Discussion)?
Conclusion: “Muscle strength expressed according to the body weight, BMI and fat mass is proved to be a sensible strategy to identify adolescent …”. Please rephrase this sentence and replace proved with i.e. “This study suggests that muscle strength...”
Introduction
Page 1: “…in view of the inverse association with adverse cardiometabolic conditions analyzed individually (e.g., body mass index (BMI), waist circumference, systolic and diastolic blood pressure, fasting blood glucose, glycated hemoglobin, total cholesterol, triglycerides, high sensitive-C-reactive protein).” Body mass index, waist circumference, systolic blood pressure etc. are not adverse cardiometabolic conditions per see. Please rephrase sentence.
Page 1-2: “Muscle strength has been described as an important health marker in adolescents [1,2], in view of the inverse association with adverse cardiometabolic conditions analyzed individually…”. “However, the beneficial effect attributed to muscle strength on adverse cardiometabolic conditions in adolescents does not seem to be a consensus, since no association or even controversial results (e.g., muscle strength directly associated with adverse cardiometabolic conditions) have been described [1–3]. And continue…” The conflicting results for the association between muscle strength and adverse cardiometabolic conditions in adolescents…”. Please rephase the second sentence to make the statement of the missing consensus in this field of research (and how this differ from the first sentence) clearer to the reader before going on describing the conflicting results.
Page 2: Please consider using absolute values of muscle strength instead of raw values.
Page 2: “…the Term of Free and Informed Consent signed by the guardians (age <18 years)…”. This reads as if the guardians themselves were under the age of 18 years. Please rephrase or add more information.
Methods
Page 3: The research team was consisting of undergraduate and graduate students. Were they the leading force in both the planning and execution of the study, or “just” the test team? Also, did the students have any pre-training before the test period for the reason of increasing standardization? If so, please add this information.
Page 3-5: Please add more details of how all the different test results were measured/recorded (i.e. anthropometrics: height was measured to the nearest i.e. 0.1 cm?).
Page 4: Regarding measuring handgrip strength, did you use 1) verbal encouragement and 2) was the dynamometer individualized/adjusted according to hand size? Please also add information if handgrip strength were used as an expression of upper body muscle strength or whole body muscle strength in the present study (including references to justify it).
Page 6: In consideration of sensitive/personal data information, under which conditions were sexual maturation measured/what considerations did you (researchers) provide for this measurement?
Page 6: “Mean and standard deviation were used to describe symmetric continuous variables, and median and interquartile range (p25–p75) for asymmetric variables.” I guess you mean “normally distributed” variables and non-normal or skewed variables?
Page 6: Please replace OR with CI95% with OR (95% CI).
Page 6: “All analyses were adjusted for all possible confounders (sociodemographic and lifestyle variables)…”. Please delete “all” since this refers to more confounders than the present study has obtained and elaborate on what is meant by sociodemographic and lifestyle variables.
Results
Page 7, 8 and 10: Please highlight significant results in Table 1, 2, and 3 (i.e. bold values or by markings (*)).
Page 8-9: Please consider adding Figure 1 and Figure 2 to the supplements and include descriptive statistics of the population in the article.
Page 9: Please add sex of participants for Figure 2
Discussion
Page 10: “…men and women” refers to the sex of adults. Please be consistent and use i.e. male and female adolescents.
Page 10: “…and inversely associated with the simultaneous presence of two or more adverse cardiometabolic conditions among female”. Can you please explain why the results from these analyses are highlighted for females only?
Page 10: The physiological cross-sectional area of the muscle(s) and a person’s ability to activate those muscles (i.e. motor unit recruitment) are important for total muscle strength. Can you elaborate on causal or correlational relationship between these factors and height that can contribute to your rationalization of why you choose to adjust handgrip strength for height?
Page 11: “…and individual components of the metabolic syndrome analyzed in a grouped way—e.g., obesity, high blood pressure, dyslipidemia, dysglycemia)” Please explain what you mean by this sentence. Do you mean clustering of these variables? Please elaborate/rephrase.
Page 11: “With regard to the restricted results to women, since women have less muscle mass and muscle quality [36] and tend to have greater difficulties in improving these components compared to men [36]”. Please justify this comparison between woman and men with regards to the study population.
Page 11: “Thus, it is hypothesized that higher levels of muscle strength may be related to greater muscle mass and muscle quality [6],…”. Please rephrase or remove “may be” in the first part of the sentence, since these association are established in the literature.
Page 11: “…was inversely associated with muscle strength expressed according to body weight (assessed by handgrip)…” It reads as body weight are assessed by handgrip. Please be careful to put in information where it is appropriate.
General comments to the discussion:
Citation from Smith, J.J. et al. (Sports Med 44, 1209–1223 (2014)):”As performance in many weight-bearing MF tests is highly correlated with body mass/adiposity (Woods et al. Pediatr Exerc Sci. 1992;4(4):302–11), analyses of the relationship between MF and the health outcome of interest should adjust for this variable in order to ascertain the independent contribution of MF”. Please add to your discussion of handgrip as a method, how it separates from weight-bearing muscle fitness/strength tests like i.e. vertical or standing long jump and push ups, and in this perspective, add more to the discussion about why you need to adjust for body mass/obesity measures when using handgrip as a measure of muscle strength (considering that it is not an weight-bearing activity).
Author Response

(The authors gave the same response as above.)

Round 2
Reviewer 1 Report
-----( General remarks )-----
The authors have made a number of corrections that enhance the value of the work. However, it still requires thorough refinement. Also, the style of the statement needs to be improved, as it has been very long-winded so far. The text of the article needs thorough rewording in order to be made more concise and clear.
If we understand this correctly, the authors have taken as the main aim of this paper to find a way for normalising muscle strength that is most useful in assessing cardiometabolic risk, that is based on estimated associations between muscle strength and cardiometabolic risk factors. If so, the value of this work does not come from stating that the measured strength value should be normalised, but first and foremost HOW. It boils down to presenting a normalisation formula that takes into account selected factors.
The results were analysed separately for both sexes, which is appropriate, but no attempt was made to include sex as a normalising factor. The need to consider this should have been expressed already at the Introduction stage, addressed in the analysis and discussed in the Discussion section. The sexes at the same metric age were assessed, which means that they are at a different stage of developmental maturity which is an additional (besides the sex factor per se) source of sex differentiation in the results obtained. Normalization of muscle strength would therefore have to take into account the sex factor and the level of developmental maturity.
The results described in this paper for the estimation of cardiometabolic risk cannot therefore be treated summarily, but be subordinated to a multidimensional assessment model that takes into account, the developmental pattern to which muscle strength is related.
-----( Title )-----
The title brings your manuscript to the readers' attention. It should therefore present in the most concise manner possible the research problem being investigated or the main finding for its solution. However, the title still does not address these issues -- it merely refers to one of the results.
The title could be as follows:
"Muscle strength as a predictor of cardiometabolic risk factors in adolescents"
"Normalisation of muscle strength measurements in the assesment of cardiometabolic risk factor"
"The value of muscle strength in the assessment of cardiometabolic risk factors should be normalised by ..."
(These examples came to mind 'at a glance'. Please work out the best solution for yourself)
-----( Abstract )-----
"Little is known about normalization [...] in the relationship with [...]"
"Normalisation" is not a problem in itself. It serves some purpose. As a background to the research problem (i.e. the muscle strength in the assessment of cardiometabolic risk factor), rather the gap in the knowledge about estimating cardiometabolic risk and the need of its fullfillment should be described. "Normalisation" in turn can be presented as a discovered method of meeting this need.
There is no need to cite numerical descriptions of research results in the abstract, as this numerical description only becomes understandable after reading the entire article. At the stage of reading the abstract, nobody will use them, and they only make reading difficult (apart from the fact that nobody, as yet, knows what OR and CI stand for). We suggest that at this point the numerical description should be abandoned in favour of a more verbal description of the result, focusing on its relevance to solving the problem and presenting the main finding and its conclusions.
-----( Introduction )-----
Lines 45-47:
We feel, that this part of the sencence:
"[...], little is known about normalization of muscle strength for body size in the relationship with cardiometabolic risk factors in adolescents. The rationale for normalizing muscle strength for body size in association with cardiometabolic risk factors is justified because [...]"
should be reworded to express the need for different ways of normalisation for different assessment methods subordinated to their research objectives.
Lines 57-62:
This sentence is poorly worded (Although [...], although [...]). It is difficult to recognise what follows from what.
One of the way it can be reworded might be i.e.: "Although some researchers have expressed the need to normalize [...] [1,2], in studies that has had synthesized the results [...] the normalisation quantities of muscle strength measurements was restricted to body mass only."
Lines 62-69:
The "is justified" at the end of the sentence is not enough. The Authors expressed the need to analyse the correlations between the values of the different assessment indices in order to develop a formula for muscle strength normalisation, but did not sufficiently explain the importance of knowing these correlations for the development of this formula (that the reduction in the influence of certain factors on the assessment of muscle strength, in terms of risk, can be achieved).
Lines 73-75:
The hypothesis is not clearly expressed. I guessed the meaning of that sentence as follows:
"We hypothesized that muscle strength normalised with body weight, BMI, height and fat mass is superior to absolute muscle strength predictor of cardiometabolic risk."
But it is only my guess (Am I right?). The Authors should express it more clear.
However it should be consider if the hypothesis results from the scientific problem stated?
-----( Materials and Methods )-----
Lines 288-289:
It is OK, but please consider possibly simplifying this notation by clarifying the format in the Materials and Methods section, e.g:
"The results were expressed as odds ratio effect size (OR) along with 95% confidence intervals (CIs),
the borders of which were presented in square brackets ([CI_lower CI_upper])"
, and present the results in this manner:
OR: 0.1 [0.05 0.2]
instead of:
OR: 0.1; 95% CI: 0.05 – 0.2
This way of writing seems more concise and clearer to us. However, this is only a suggestion from our side - we do not expect the authors to necessarily follow it.
Diagram: insert ALL indices, including those relating to risk assessment, and illustrate their function in assessment of the risk.
-----( Results )-----
Line 305:
The minor remark: What does the "±" mean in "adolescents (age, 16.6 ± 1.0 years"? Standard deviation? Range? Please make it clear.
Figure 2, Figure 3:
The order of colors of approximation lines in diagrams A and B a is opposite to that in diagrams B, C, and D. Correcting this inconsistency will make the diagram easier to understand.
Tables:
The headings of the tables contain the names of the units of measurements. In the context of the OR values presented in the table, relating to the relationship, these names are not only irrelevant but may be misleading.
We feel, that presenting a log odds (logit transformed OR) instead of raw OR would be much easier to understand and interpret (0 means "no association" and the sign of the number indicates its direction, if the 95%CI borders have reverse signs, the p-value is probably > 0.05). Another advantage is linearisation of the scale of interpretation, useful in e.g. comparisons between different values of the effect size.
-----( Discussion )-----
Discussion contains now a lot of very important information, which can be more helpful in the introduction, so they should be shifted rather into the introduction section.
The interpretation of results is lacking.
It is not stated what is the main finding. The partial results are discussed which do not lead to the final conclusion.
-----( Conclusions )-----
The conclusions are concerned also with partial results and not refer to scientific problem and hypothesis posed.
Author Response
Dear reviewer,
We thank you once again for your careful reading and relevant comments regarding the manuscript submitted for review. We are also grateful for the opportunity to once again be able to answer the questions raised, and in this way, present an improved version of the manuscript.
Best regards,
Author's!

Reviewer 3 Report
I think the authors have done a great job during the revision. I do, however, still think the manuscript could benefit from a last round of text editing (with special attention to shortening/rephrasing the longest paragraphs).
Author Response
Dear reviewer,
We are grateful for the comments made by the reviewer, and, in order to respond to the suggestions made, an additional effort was made regarding the editing of the contents inserted in the manuscript.
Best regards,
Author's!
